# Nanotechnology-Based Delivery of CRISPR/Cas9 for Cancer Treatment: A Comprehensive Review

**DOI:** 10.3390/cells14151136

**Published:** 2025-07-23

**Authors:** Mohd Ahmar Rauf, Afifa Rao, Siva Sankari Sivasoorian, Arun K. Iyer

**Affiliations:** 1Department of Internal Medicine, Heme/Oncology Unit, University of Michigan, Ann Arbor, MI 48109, USA; 2Department of Pharmaceutical Sciences, Eugene Applebaum College of Pharmacy and Health Sciences, Wayne State University, Detroit, MI 48201, USA; hv3814@wayne.edu; 3Jamia Hamdard Medical College, Jamia Hamdard University, New Delhi 110062, India; afeefa.rao780@gmail.com

**Keywords:** nanotechnology, CRISPR/Cas9, gene editing

## Abstract

CRISPR/Cas9 (Clustered Regularly Interspaced Short Palindromic Repeats-associated protein 9)-mediated genome editing has emerged as a transformative tool in medicine, offering significant potential for cancer therapy because of its capacity to precisely target and alter the genetic modifications associated with the disease. However, a major challenge for its clinical translation is the safe and efficient in vivo delivery of CRISPR/Cas9 components to target cells. Nanotechnology is a promising solution to this problem. Nanocarriers, owing to their tunable physicochemical properties, can encapsulate and protect CRISPR/Cas9 components, enabling targeted delivery and enhanced cellular uptake. This review provides a comprehensive examination of the synergistic potential of CRISPR/Cas9 and nanotechnology in cancer therapy and explores their integrated therapeutic applications in gene editing and immunotherapy. A critical aspect of in vivo CRISPR/Cas9 application is to achieve effective localization at the tumor site while minimizing off-target effects. Nanocarriers can be engineered to overcome biological barriers, thereby augmenting tumor-specific delivery and facilitating intracellular uptake. Furthermore, their design allows for controlled release of the therapeutic payload, ensuring sustained efficacy and reduced systemic toxicity. The optimization of nanocarrier attributes, including size, shape, surface charge, and composition, is crucial for improving the cellular internalization, endosomal escape, and nuclear localization of CRISPR/Cas9. Moreover, surface functionalization with targeting ligands can enhance the specificity of cancer cells, leading to improved gene-editing accuracy. This review thoroughly discusses the challenges associated with in vivo CRISPR/Cas9 delivery and the innovative nanotechnological strategies employed to overcome them, highlighting their combined potential for advancing cancer treatment for clinical application.

## 1. Introduction

Cancer continues to pose a significant threat to human health, frequently leading to elevated mortality rates despite progress in standard therapies such as chemotherapy, radiation, and surgical interventions [1,2]. The persistent emergence of cancer and evolution of resistance to conventional treatments highlight the need for innovative therapeutic strategies. The known significance of genetic and epigenetic abnormalities in cancer formation and progression, with more than 600 mutations recorded in the catalog of somatic abnormalities in cancer, underscores the possibility of precise gene editing as an effective therapeutic approach [3,4,5].

The CRISPR-Cas9 system has been developed into a multifaceted and effective gene-editing instrument that is proficient in implementing precise alterations to the genome. This method employs a single guide RNA (sgRNA) to route the Cas9 endonuclease to a specific genomic location, facilitating accurate DNA cleavage and consequent gene disruption or repair [6,7]. In comparison to previous gene editing methods, such as ZFNs and TALENs, CRISPR-Cas9 provides enhanced flexibility, efficiency, and precision, positioning it as a promising option for therapeutic applications [6,8,9]. Since its inaugural use in mammalian cells in 2013, CRISPR-Cas9 has rapidly garnered prominence across multiple domains, including oncological research. The ability to modify tumor-associated genes, improve tumor immunotherapy, enable disease modeling, and collaborate with traditional anticancer medications has garnered considerable interest among scientists [10,11].

The effective implementation of CRISPR-Cas9 technology in clinical practice depends on the safe and efficient delivery of its components to the target cells. Cas9 protein and sgRNA must be efficiently delivered to the nucleus for accurate gene editing. Although physical and viral delivery strategies have been investigated, they frequently encounter constraints including low efficiency, immunogenicity, restricted packaging capacity, and off-target effects [12,13,14]. Nanotechnology is a viable solution to this issue, utilizing materials on a nanoscale that exhibit unique physicochemical characteristics enabling efficient encapsulation and safeguarding of CRISPR/Cas9 components, thereby promoting their targeted transport and cellular uptake. A variety of nanomaterials, such as polymers, lipids, porous silicon, mesoporous silica nanoparticles, and metal–organic frameworks, have been examined for this purpose. This review examines the synergistic potential of CRISPR/Cas9 and nanotechnology in cancer therapy and investigates the various therapeutic applications of this integrated method in gene editing and immunotherapy [12,15,16,17,18].

This review examines the effective CRISPR-Cas9 delivery and its use in cancer treatment. We explored its significance in augmenting the in vivo distribution, strengthening CRISPR-Cas9 effectiveness, and supporting cancer immunotherapy. Rapid progress in nanotechnology-based delivery methods offers significant potential for enhancing the therapeutic capabilities of CRISPR-Cas9 technology and tackling intricate issues in cancer treatment (Figure 1).

## 2. From Bacterial Defense to Gene Editing Revolution: The CRISPR-Cas9 Story

The innovative CRISPR-Cas9 gene editing technology originates from an ancient bacterial defense mechanism. Microbes that are perpetually endangered by invading viruses have developed an extraordinary adaptive immune system for their protection. The clustered regularly interspaced short palindromic repeats (CRISPR) system operates by seizing fragments of viral DNA and preserving them as “memories” in the bacterial genome. DNA fragments are translated into short RNA molecules that direct an enzyme known as Cas to accurately identify and eliminate the relevant viral DNA during subsequent infections. Bacteria retain memories of previous intruders and employ this information to execute a precise response. Scientists acknowledging the system’s promise promptly modified it for gene editing applications. In a few years, they effectively utilized the CRISPR-Cas system to alter DNA sequences in diverse organisms, beginning with bacteria and ultimately achieving the precise editing of human cells. This advancement has initiated a novel epoch of genetic manipulation, providing unparalleled control over the genetic composition of living beings [20,21,22,23].

## 3. CRISPR-Cas9: A Bacterial Defense Mechanism Repurposed for Accurate Gene Editing

The CRISPR-Cas9 system, an innovative gene editing tool, originated from a bacterial defensive mechanism first identified in Escherichia coli in 1987 [24,25]. CRISPR systems, categorized into two primary categories with 16 subtypes according to their endonuclease recognition and cleavage methods, confer adaptive immunity against intruding genetic elements, such as bacteriophages and plasmids. The type II CRISPR system, notably CRISPR-Cas9, has become prominent for its capacity to precisely and efficiently target and change specific DNA regions, especially in mammalian cells [26,27,28].

The CRISPR-Cas9 system comprises the following three essential components: CRISPR RNA (crRNA), trans-activating CRISPR RNA (tracrRNA), and Cas9 endonuclease. The crRNA consists of an additional area that hybridizes with tracrRNA by complementary base pairing, as well as a 20 nucleotide spacer sequence that guides the Cas9 protein to the target DNA locus. A ribonucleoprotein (RNP) complex is created when TracrRNA attaches itself to the Cas9 protein. It is common practice to combine crRNAs and tracrRNAs into single guide RNA (sgRNA) in order to simplify the mechanism and increase target DNA affinity (Figure 2). Target recognition entails the identification of a specific DNA sequence by crRNA, whereas the Cas9 enzyme detects a short sequence pattern referred to as the protospacer adjacent motif (PAM), generally situated downstream of the target site. The PAM sequence functions as an essential recognition component, differentiating the host’s DNA from exogenous DNA and averting autoimmunity [29,30,31].

The Cas9 endonuclease comprises six principal domains, including the HNH and RuvC domains tasked with cleaving the target DNA [32,33,34]. The HNH domain cleaves the DNA strand complementary to the crRNA, whereas the RuvC domain cleaves the non-complementary strand. Subsequent to DNA cleavage, cellular DNA repair processes are activated [35,36]. Non-homologous end joining (NHEJ) is an error-prone repair mechanism that may result in insertions or deletions (indels) at the cleavage site, causing gene disruption or knockout. Conversely, homology-directed repair (HDR) employs a donor DNA template for accurate gene editing, facilitating the incorporation of alterations. Nonetheless, HDR is less effective than NHEJ and is limited to proliferating cells, resulting in a lower efficiency of gene knock-in compared to deletion [37,38].

**Figure 2 cells-14-01136-f002:**
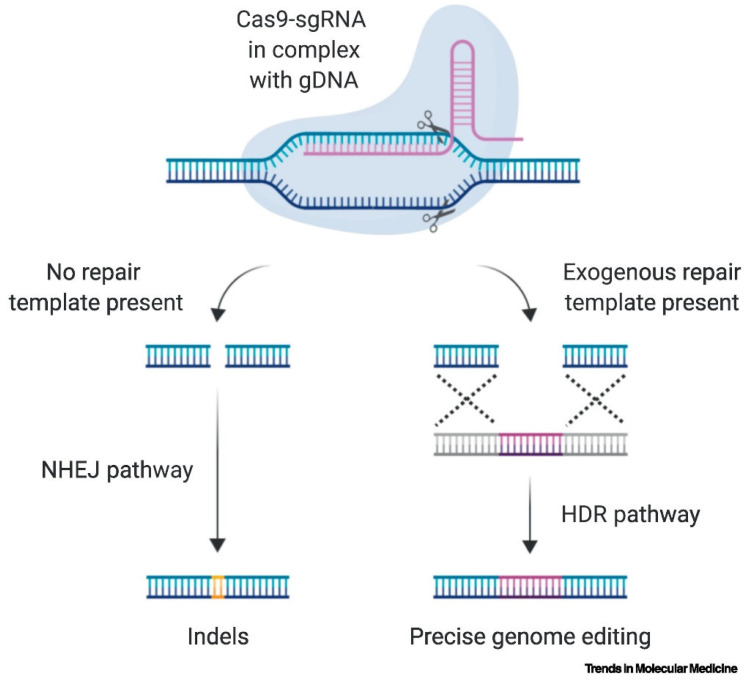
Mechanisms to repair CRISPR-induced DSBs. Adapted from Kampen et al. 2019 [39] with permission.

## 4. Nanotechnology-Based Delivery of CRISPR/Cas9 for Cancer Treatment

### 4.1. Lipid Nanoparticles (LNPs) for CRISPR/Cas9 Plasmid Delivery

Lipid nanoparticles (LNPs) have attracted considerable interest as effective media for nucleic acid delivery, especially for CRISPR/Cas9 gene editing. The fundamental mechanism of LNP-mediated transport is based on the electrostatic interactions between cationic lipids and DNA’s negatively charged phosphate backbone of DNA. This connection enables the encapsulation of plasmid DNA within the LNP core and enhances cellular absorption by engaging the negatively charged cell membrane. Comprehensive research has resulted in the development of various LNP formulations for delivering CRISPR/Cas9 plasmids. Wang et al. 2022 [40] created a multifunctional lipid nanoparticle (LNP) to deliver a plasmid containing Cas9 targeting the MutT homolog 1 (MTH1) gene in non-small cell lung cancer (NSCLC) cells. This advanced LNP design entailed pre-condensing the plasmid with protamine, a positively charged protein containing a nuclear localization sequence (NLS) and subsequent encapsulation into a cationic liposome. Further modifications with DSPE-PEG-HA endowed the nanocarriers with tumor-targeting capabilities, leading to enhanced cellular internalization and nuclear localization. This improved delivery resulted in effective MTH1 gene disruption and subsequent suppression of NSCLC development [40]. 

LNP formulations have been customized to fulfill diverse therapeutic goals such as targeted distribution, extended circulation duration, increased transfection efficiency, less toxicity, and greater particle stability. Liu et al.2018. [41] developed cationic lipid-assisted polymeric nanoparticles (CLANs) for the targeted delivery of a Cas9 plasmid aimed at the BCR-ABL fusion gene in chronic myeloid leukemia (CML) cells. The CLANs, consisting of a cationic lipid (BHEM-Chol) and a PEG-PLGA polymer, facilitated accurate gene editing of the BCR-ABL fusion gene, resulting in an extended longevity and less leukemia load in CML mice [41]. To improve the treatment efficacy, Chen et al.2022. [42] created a multifunctional cationic lipid nanoparticle (DOX-CB@lipo-pDNA-iRGD) that co-delivered a Cas9 plasmid targeting the CD47 gene and a boron compound. This lipid nanoparticle (LNP) included doxorubicin (DOX) owing to its nuclear affinity and the iRGD peptide for tumor localization. This method integrated CRISPR/Cas9-mediated CD47 disruption with boron neutron capture therapy (BNCT), leading to enhanced antitumor efficacy and increased survival rates in tumor-bearing mice. Recent progress in LNP design includes the creation of ionizable LNPs (iLNPs) that exhibit improved endosomal escape characteristics. Li et al. 2022. [43] documented an iLNP (iLP181) that effectively delivered a Cas9 plasmid aimed at polo-like kinase 1 (PLK1) in hepatoma carcinoma cells. This iLNP demonstrated enhanced endosomal escape and gene editing efficacy relative to commercial Lipofectamine 2000, resulting in substantial tumor growth suppression in vivo. In addition to enhancing delivery and therapeutic efficiency, LNPs provide the benefit of safeguarding plasmid DNA against degradation and immunological reactions. In addition, LNPs can be easily altered by targeting ligands or other functional groups to augment their therapeutic efficacy. These characteristics underscore the adaptability and potential of LNPs as a secure and efficient medium for CRISPR/Cas9 plasmid delivery in oncological treatment [42].

### 4.2. Inorganic and Cationic Nanomaterials for CRISPR/Cas9 Plasmid Delivery

In addition to polymeric nanoparticles, inorganic nanomaterials have emerged as promising options for pCas9 delivery, owing to their distinctive characteristics that enhance cellular absorption and endosomal release. Metal–organic frameworks (MOFs) have attracted significant interest as potential platforms for biological applications. Metal–organic frameworks (MOFs) consist of metal ions or clusters coupled with organic ligands, resulting in porous structures with adjustable characteristics [44].

The investigation of MOFs for pCas9 delivery is still in its early phases; however, some encouraging outcomes have been documented. A recent study illustrated the application of polymer-functionalized NH2-UiO-66 MOF for the co-delivery of doxorubicin (DOX) and pCas9. Despite the transfection efficiency being moderate, this study underscores the potential of MOFs as carriers for CRISPR/Cas9 components [45].

Zeolitic imidazole frameworks (ZIFs), a subset of metal–organic frameworks (MOFs), exhibit significant potential for the delivery of biological macromolecules. ZIF-8 has been thoroughly studied for its capacity to transport functional genes. Pyreddy et al. 2023 [46] innovated a ZIF-based platform (ZIF-C) for the delivery of pCas9, attaining notable gene editing efficacy in prostate cancer cells. Poddar et al.2019 [47] recently illustrated the efficacy of a pCas9-ZIF-8 system for Paxillin knock-in through homology-directed repair (HDR), highlighting the adaptability of ZIFs for intricate gene editing endeavors.

Gold nanoparticles (AuNPs) constitute a compelling method for the administration of pCas9. Chen et. al.2021. [48] conducted a thorough investigation into the impact of the aspect ratio of Au nanorods (AuNR) on pCas9 delivery and gene editing efficacy. Their research demonstrated that high-aspect-ratio AuNRs, treated with cationic polymers such as PEI, displayed enhanced DNA binding, cellular uptake, and endosomal escape, resulting in effective gene editing. The same group subsequently advanced supramolecular cationic AuNRs that integrated NIR-II light-activated hyperthermia with CRISPR/Cas9 delivery to improve gene editing and induce immunogenic cell death [49].

Hollow mesoporous silica nanoparticles (HMSNs) represent a distinct category of inorganic nanomaterials with appealing characteristics for gene delivery. Their distinctive hollow mesoporous architecture facilitates the encapsulation of medicines and gene editing elements. A recent work illustrated the co-delivery of sorafenib and pCas9 via polyamidoamine-aptamer-coated HMSNs, resulting in effective gene knockdown and synergistic therapeutic outcomes in liver cancer cells [50].

Cationic polymers, owing to their capacity to condense and encapsulate negatively charged DNA, have become fundamental in the development of nanocarriers for pCas9 distribution. These polymers utilize electrostatic interactions to create complexes with plasmid DNA, thereby enhancing their effective packing and intracellular delivery. Protamine, a naturally occurring cationic protein, has been extensively used for DNA condensation. He et al.2019. [51] used protamine along with calcium carbonate (CaCO3) to build a core for encapsulating pCas9 aimed at β-catenin. The core was subsequently coated with hyaluronic acid modified by the TAT-NLS peptide and AS1411 aptamer to improve cellular absorption and nuclear targeting. The resultant nanocarrier effectively accomplished β-catenin knockdown owing to the combined effects of protamine-facilitated DNA condensation and targeted delivery [52]. Additionally, Tao et al. 2021 [53] innovated the application of protamine-capped gold nanoclusters (protamine-AuNCs) as a delivery vehicle for pCas9, leveraging the benefits of protamine’s cationic characteristics and the biocompatibility and cellular absorption capabilities of AuNCs.

Chitosan, a naturally sourced polysaccharide rich in amino groups, is a promising candidate for pCas9 delivery. Their facile modification and biocompatibility have catalyzed the advancement of diverse chitosan-based nanocarriers [54]. A nanoplatform comprising folic acid (FA) and 2-(diisopropylamino) ethyl methacrylate double-grafted trimethyl chitosan (TMC) was created for the concurrent delivery of a plasmid expressing Survivin (sgSurvivin pDNA) and DOX. This approach efficiently condensed and safeguarded plasmid DNA, facilitating successful gene editing and enhanced anticancer efficacy both in vitro and in vivo. Shao et al. created a lactobionic acid-modified chitosan nanocarrier for the targeted delivery of PTX and sg-VEGFR2/Cas9 plasmid to hepatocellular carcinoma (HCC) cells, resulting in substantial tumor growth suppression [55].

Dendrimers, characterized by their well-structured branching architecture and elevated density of surface functional groups, provide distinct benefits for gene delivery [56,57]. Yang et al. engineered a fluorinated G5 PAMAM dendrimer (F-PC) to encapsulate HSP70-promoter-driven pCas9 targeting PD-L1 and photosensitizer Ce6. This approach integrates photodynamic treatment with CRISPR/Cas9-mediated gene editing to improve antitumor efficacy. The HSP70 promoter facilitated spatial regulation of Cas9 expression, hence reducing possible off-target effects [55].

Poly (β-amino ester) (PBAE) is a flexible cationic polymer that is extensively investigated for gene delivery applications. Advances in Polymer engineering have led to multifunctional linear- branched and hyperbranched PBAEs that demonstrate superior gene transfection and silencing efficacy compared to conventional reagents such as PEI. Moreover, Zhang et al. 2023 [58] developed an internally and externally engineered exosome (IEEE) by encapsulating a PBAE/CRISPRi/dCas9 nanocomplex within CRV-expressing exosome membranes for precise delivery to tumor-associated macrophages (TAMs), thereby repolarizing them from an M2 to an M1 phenotype and suppressing tumor growth [59,60]. 

Poly (disulfide) (PD) has recently garnered interest owing to its capacity to enable intracellular cargo transit through disulfide exchange mechanisms, circumventing endosomal trapping. Xu et al. 2022 [61] demonstrated the effective delivery of plasmids with a PEG-modified PD, whereas Ping et al. 2022 [61] employed PD for the liver-targeted administration of a Cas-editing plasmid.

### 4.3. Barriers to CRISPR/Cas9 Delivery Systems: mRNA-Based CRISPR/Cas9 Systems

The RNA-based CRISPR/Cas9 system employing Cas9 mRNA and sgRNA offers a persuasive alternative to plasmid-based methodologies. Their ephemeral characteristics, diminished likelihood of off-target effects, and reduced immunogenicity provide considerable benefits for their therapeutic use. Nonetheless, effective administration of mRNA continues to pose a significant challenge. The intrinsic instability of mRNA and its restricted cellular permeability require the development of effective delivery methods.

Although mRNA translation occurs in the cytoplasm, eliminating the necessity for nuclear entrance, the considerable bulk and vulnerability to Cas9 mRNA degradation present obstacles for effective delivery. Moreover, effective gene editing necessitates concurrent introduction of both Cas9 mRNA and sgRNA into identical cells. To address these issues, numerous nanocarrier systems have been examined, with lipid nanoparticles (LNPs) being the most extensively studied. Liu et al. 2019 [62] created bioreducible lipid nanoparticles (LNPs) using a new cationic lipid, BAMEA-O16B, for the simultaneous delivery of Cas9 mRNA and sgRNA. These LNPs efficiently contained and safeguarded mRNA cargo, which was released into the cytoplasm upon exposure to reductive stimuli. This method accomplished exceptionally rapid and effective gene editing, both in vitro and in vivo. Rosenblum et al. 2020 [63] developed a library of amino ionizable lipids to create lipid nanoparticles for the delivery of chemically modified Cas9 mRNA and sgRNA. These alterations improve mRNA stability and decrease immunogenicity. Solitary intracerebral delivery of these lipid nanoparticles aimed at PLK1 in an orthotopic glioma model led to considerable tumor growth suppression and enhanced survival rates. Endosomal escape continues to be a significant obstacle in mRNA distribution. Siegwart et al. created a series of innovative phospholipids (iPhos) that exhibited improved endosomal escape characteristics. iPhos establishes a conical configuration in the acidic endosomal milieu, promoting membrane breakdown and cargo liberation. The resultant iPhos-based lipid nanoparticles (iPLNPs) facilitated effective mRNA transport and CRISPR/Cas9-mediated gene editing in vivo. Farbiak et al. 2021 [64] conducted sophisticated work in which they developed dendrimer-based lipid nanoparticles (dLNPs) for the delivery of a comprehensive nucleic acid CRISPR/Cas9 platform, including Cas9 mRNA, sgRNA, and donor ssDNA. These dLNPs demonstrated exceptional HDR efficiency both in vitro and in vivo, underscoring their promise for gene repair applications. Zhang et al. 2022 [65] employed dLNPs to simultaneously deliver siRNA targeting focal adhesion kinase (FAK) alongside Cas9 mRNA and sgRNA, therefore improving tumor targeting and gene editing efficacy. This method markedly boosted dLNP absorption and tumor infiltration, resulting in superior gene editing and tumor growth suppression in many cancer models. In addition to lipid nanoparticles, alternative nanocarrier platforms have also been investigated for mRNA delivery. Abbasi et al. 2021 [66] engineered PEGylated polyplex micelles (PMs) for simultaneous delivery of Cas9 mRNA and sgRNA. These PMs efficiently encapsulated and safeguarded mRNA payload, improved sgRNA stability, and promoted endosomal release. Extracellular vesicles (EVs) have demonstrated the potential for mRNA delivery. Le et al. 2018 [67] employed red blood cell-derived extracellular vesicles (RBCEVs) to deliver an mRNA-based CRISPR/Cas9 system, resulting in significant gene editing efficacy and reduced toxicity. Li et al. 2019 [68] enhanced RNA loading efficiency by engineering exosomes with an RNA-binding protein (HuR) linked to the exosome membrane protein CD9, facilitating the effective encapsulation of ARE-modified Cas9 mRNA.

### 4.4. RNP-Based CRISPR/Cas9: Direct Administration for Improved Gene Editing

Direct administration of the CRISPR/Cas9 system as a ribonucleoprotein (RNP) complex, consisting of Cas9 protein and sgRNA, constitutes a simplified and effective approach for gene editing. This approach bypasses intracellular transcription and translation and presents multiple advantages over plasmid- or mRNA-based techniques. These benefits include rapid action, reduction in off-target effects, and mitigation of cytotoxicity. Nonetheless, the administration of RNPs poses distinct issues owing to their complex structures and charge characteristics. The positively charged Cas9 protein, when complexed with negatively charged sgRNA, yields an overall net negative charge for RNP, impeding its capacity to traverse the cell membrane efficiently. To address these delivery challenges, researchers have focused on creating a variety of nanocarriers specifically engineered for RNP delivery. These nanocarriers enclose and safeguard the RNP complex, promoting cellular uptake through endocytosis and subsequent release into the cytoplasm via endosomal escape. Upon entering the nucleus, RNPs can perform their gene editing function. Contemporary nanocarrier methodologies for RNP’s distribution can be roughly classified into the following four primary categories: lipid-, polymer-, inorganic-, and cell membrane-based.

### 4.5. Lipid-Based Nanotechnology for Ribonucleoprotein Delivery

Lipid nanoparticles (LNPs) have attracted considerable interest as a viable platform for RNP administration owing to their proven efficacy in delivering genes and proteins both in vitro and in vivo. Notably, one of the three CRISPR-based therapies currently approved for clinical trials targeting transthyretin amyloidosis utilizes LNP-mediated delivery. LNPs have also demonstrated high gene editing efficiency in neuronal systems, particularly in ex vivo applications involving cortical neurons. Additionally, microfluidic platforms have been engineered to produce RNP loaded LNPs with enhanced contro, achieving efficient gene editing while minimizing toxicity. Additionally, an FDA-sanctioned LNP formulation (DLin-MC3-DMA) has been effectively utilized to co-deliver RNP targeting NFE2L2 alongside a sonosensitizer to augment sonodynamic therapy. Walther et al. methodically adjusted the lipid nanoparticle formulation parameters for ribonucleoprotein distribution, emphasizing the significance of buffer composition and cationic lipid concentration in attaining effective gene knockout and correction. Their research emphasized the necessity for customized LNP formulations based on the particular gene editing application [69]. In another study, Qiang et al. demonstrated an advanced, versatile lipid nanoparticle platform for the effective delivery of ribonucleoproteins to many cell types and tissues, based on prior research. The incorporation of the cationic lipid DOTAP into the LNP formulation improved RNP encapsulation and gene editing efficiency above 80%. This platform offers considerable potential for both in vitro and in vivo gene editing applications [70] [Figure 3].

CRISPR/Cas9 ribonucleoprotein (RNP) systems have been delivered using strategies that enhance targeting and precision. One approach involves encapsulating RNPs within nanofibrils coated with mesenchymal stem cell membranes and integrated with chemoattractants to improve accumulation and retention in the bone marrow niche leading to effective gene editing in hematological malignancies. Cai et al. 2022 [71] recently employed biodegradable LNPs to administer an enhanced inactivated CRISPR (eiCRISPR) system to a xenograft tumor model. This approach, consisting of Cas9 protein, an inactive self-contained sgRNA, and a chemically caged DNAzyme, facilitated the regulated activation of gene editing exclusively within the tumor microenvironment, thereby improving both safety and efficacy.

### 4.6. Polymer Nanocarriers for Improved CRISPR/Cas9 RNP Delivery

Polymer-based nanocarriers have emerged as a viable method for delivering CRISPR/Cas9 ribonucleoprotein (RNP) complexes, including benefits such as colloidal stability, biocompatibility, and capacity for customization for targeted distribution and cell membrane penetration. Beyond conventional carriers such as polyethyleneimine (PEI), recent approaches include chemically modified RNP systems with improved stability and gene editing activity. DNA nanostructures are also being explored for their ability to encapsulate RNPs via sequence-specific recognition motifs, offering enhanced protection and controlled release in targeted cell populations. Other strategies leverage functionalized polymers to achieve tumor-specific delivery and immune checkpoint disruption. Notably, Ruan et al. 2022 [72] produced a polymer modified with angiopep-2 coated with guanidinium and fluorine for targeted RNP delivery to the brain. This nanocarrier successfully traversed the blood–brain barrier and accomplished substantial gene deletion and protein decrease in glioblastoma models. Separately, DNA nanoflower platforms incorporating MUC1 aptamers and miR-21-sensitive sequences offer tumor-specific delivery and responsive gene editing [Figure 4].

### 4.7. Inorganic Nanomaterials for Ribonucleoprotein Delivery

Inorganic nanoparticles provide an effective substrate for improving the delivery of CRISPR/Cas9 RNP complexes. Their intrinsic benefits, including simple synthesis, structural stability, and varied compositional functions, provide efficient RNP loading and safeguards against degradation. Metal–organic frameworks (MOFs) have emerged as promising options for RNP delivery because of their adjustable structures, biocompatibility, and biodegradability [Figure 5] [74].

Cell-specific delivery of CRISPR-Cas9 ribonucleoprotein has been advanced using metal-organic frameworks (MOFs), such as ZIF-8, modified with biomimetic cancer cell membranes. These systems enhance tumor targeting and improve intracellular delivery efficiency. In some designs, ZIF-8 has been co-loaded with Cas9 RNPs targeting immunosuppressive pathways alongside sonosensitizers to facilitate ultrasound-triggered endosomal escape and improve gene editing outcomes. Pu et al. 2021 [75] established a sono-controllable and ROS-responsive MOF platform to facilitate the on-demand release of RNP. By covalently bonding RNP to porphyrin-integrated nanoMOFs via an ROS-sensitive linker, they achieved spatiotemporal regulation of gene editing by ultrasound stimulation. This novel method emphasizes the potential of combining sonodynamic therapy with CRISPR/Cas9 technology for improved therapeutic results.

Gold nanoparticles (AuNPs) have garnered considerable interest for RNP delivery owing to their adaptable surface modification properties and intrinsic photothermal conversion efficacy in the near-infrared (NIR) spectrum [76,77]. Functionalized gold nanorods (GNRs) and [31] multi-branched nanostructures have been designed for targeted delivery, NIR-triggered release, and synergistic photothermal gene therapy. One such design illustrated in (Figure 5) features a branched gold nanoarchitecture embedded in a mesoporous coating, co-delivering RNPs and photothermal agents for enhanced tumor ablation. Various platforms have also incorporated aptamer ligands, tumor-derived membrane coatings, or co-loaded chemotherapeutics to improve specificity and efficacy. [31] To address tumor thermoresistance, Li et al. 2021 [78] developed hypoxia-responsive AuNRs for the regulated release of RNP targeting HSP90α. This method specifically reduces HSP90α in the hypoxic tumor microenvironment, thereby increasing tumor susceptibility to photothermal therapy. Separately, copper sulfide (CuS) nanoparticles have been used for co-delivery of gene editors and chemotherapeutics or immunomodulators, supporting multimodal approaches that combine gene editing, chemotherapy, and immune activation to produce synergistic anti-tumor effects [79].

Silica nanostructures have been explored due to their structural plasticity, stability, and biocompatibility. Kim et al. 2023 [80] developed a sponge-like silica nanostructure for the effective transport of both Cas9 nuclease RNP and base editor RNP, resulting in elevated gene editing efficiency with reduced off-target effects, as shown in Figure 6 (Kim et al. 2023) [80]. Li et al. 2022 [81] constructed a cascade nanoreactor by encapsulating silica spheres with CaCO3 and including H2O2-responsive manganese carbonyl, glucose oxidase, and GSH-sensitive Cas9 RNP aimed at Nrf2. This approach utilized the tumor microenvironment for the regulated release of RNP and augmented ROS-induced tumor cell apoptosis [81].

### 4.8. Extracellular Vesicles and Polymeric Micelles for Ribonucleoprotein Delivery

In addition to the previously mentioned nanomaterials, extracellular vesicles (EVs) and polymeric micelles present distinct benefits in the delivery of CRISPR/Cas9 RNP complexes.

Extracellular vesicles (EVs), which are naturally occurring membrane vesicles secreted by cells, are essential for intercellular communication because they facilitate the transport of bioactive molecules and genetic information. Their biocompatibility, capacity to traverse biological barriers, and inherent targeting abilities render them suitable candidates for drug delivery, particularly for RNP complexes [82,83,84].

RNPs can be loaded into EVs using several strategies, including the fusion of Cas9 with EV scaffolding proteins. Cas9 fused with a WW domain (WW-Cas9) can be efficiently loaded into extracellular vehicles (EVs) via specific interactions with ARRDC1 on the EV membrane. This method has effectively facilitated RNP targeting of GFP, showcasing efficient gene knockout in recipient cells [85]. One strategy to improve RNP loading into exosomes involves modifying membrane proteins to enhance cargo recognition and encapsulation. Such approaches have been shown to increase the precision and efficiency of Cas9 RNP delivery, resulting in improved gene knockout outcomes.

Researchers have investigated methods to improve the targeting specificity of extracellular vesicles for ribonucleoprotein delivery. Targeted delivery of ribonucleoproteins(RNPs) has been achieved by engineering extracellular vesicles (Evs) with DNA aptamers and valency-controlled tetrahedral DNA nanostructures (TDNs). These modifications enhance tumor-specific accumulation of Evs and significantly improve gene editing efficiency, resulting in marked tumor growth suppression. Polymeric micelles are self-assembled nanostructures that are formed from amphiphilic polymers. They provide numerous benefits for drug delivery, such as straightforward functionalization, enhanced stability, and biocompatibility. These characteristics make them appealing for the administration of RNP complexes. Zhang et al. 2022 [86] created a novel mesoporous nanoflower platform (Cas9-NF) by cross-linking Cas9 RNP with Pluronic 127 micelles using a reduction-responsive linker. This platform facilitated effective intracellular delivery and regulated the release of Cas9 RNP, leading to significant gene editing and inhibition of tumor growth. Furthermore, the Cas9-NF nanoflowers may function as a multifunctional drug delivery system, enabling the simultaneous delivery of hydrophilic and hydrophobic drugs, thus promoting the combination of gene therapy with additional therapeutic approaches.

## 5. Clinical Applications

Gene-editing technologies, especially CRISPR/Cas9, are progressing swiftly, with multiple clinical trials investigating their therapeutic potential in different cancer types. These trials concentrate on modifying essential immune regulatory genes, including PD-1, TCR, CISH, and TRAC, to improve the effectiveness of T cell-based therapies, especially in solid tumors and hematological malignancies. A notable approach includes knockout of the PD-1 gene to improve the immune response to cancer cells. PD-1 knockout T cells are currently under investigation in adult solid tumors (NCT03545815), hormone-refractory prostate cancer (HRPC, NCT02867345), and non-small cell lung cancer (NSCLC, NCT02793856). These studies indicate that delivery methods, such as electroporation, enhance the efficiency of gene editing and are frequently utilized. The objective is to inhibit immune checkpoints that cancers utilize to avoid immune recognition, as exemplified by mesothelin-directed CAR-T cell therapies aimed at solid tumors (NCT03747965).

Furthermore, advanced CAR-T cell therapies are progressing, especially for B-cell malignancies. CTX110, a CRISPR-edited CAR-T cell therapy aimed at TRAC and β2M, is currently being studied for B-cell malignancies (NCT04035434) and shows promising recruitment and advancement in the trial. CTX130, an anti-CD70 CAR-T therapy, is under development for T-cell lymphoma (NCT04502446) and renal cell carcinoma (RCC, NCT04438083), employing electroporation for ex vivo gene editing. These therapies provide potential solutions for managing relapsed and refractory cancer types that are resistant to standard treatment.

CRISPR-based gene editing is currently under investigation for its potential application in viral-associated cancers. CRISPR/Cas9 has been utilized to target E6/E7 genes linked to human papillomavirus (HPV) in HPV-related malignancies (NCT03057912). Additionally, Cas9 plasmid-based editing has been applied to alter T cells in Epstein–Barr virus (EBV)-associated malignancies (NCT03044743). The trials demonstrated the adaptability of CRISPR technologies across various cancer types and viral origins, indicating potential wider applications in immuno-oncology.

Hematological malignancies such as acute lymphoblastic leukemia (ALL) and acute myeloid leukemia (AML) represent significant targets for CRISPR/Cas9-based therapies. Research is ongoing on CD33-deleted hematopoietic stem and progenitor cell (HSPC) therapies for acute myeloid leukemia (AML) (NCT05662904). Additionally, other strategies are being investigated for B-cell lymphoma utilizing CTX112 (NCT05643742) and BCMA-targeted therapies, such as CB-011 for relapsed or refractory multiple myeloma (NCT05722418). These treatments typically utilize ex vivo gene editing to generate modified T cells or hematopoietic cells that can elicit a more robust anticancer response. Universal CAR-T cells targeting CD19, CD20, or CD22 are under investigation for leukemia and lymphoma utilizing Cas9 mRNA and RNP methodologies (NCT03398967). This allogeneic strategy may offer a scalable resolution to the existing constraints of autologous CAR-T cell therapies. Although CRISPR-based therapies have significant potential, several challenges still persist. Multiple trials have been terminated or withdrawn owing to unexpected complications, including the trial focused on Wilms tumor 1 (WT1) utilizing Cas9 RNP (NCT05066165) and the study of PD-1 knockout T cells in metastatic RCC (NCT02867332). These setbacks highlight the necessity for ongoing improvement of gene-editing technologies, which encompasses the development of more efficient delivery systems, reduction in off-target effects, and assurance of long-term safety.

A diverse array of clinical trials employing CRISPR/Cas9 has demonstrated the growing application of gene-editing therapies in cancer treatment. These trials encompass a wide array of targets, including PD-1, TCR, β2M, and CD33, addressing various cancer types, such as solid tumors, leukemia, and lymphoma. Delivery methods, including electroporation, Cas9 mRNA, and Cas9 RNP, have demonstrated effectiveness, with numerous trials advancing to phase 1/2, indicating safety and preliminary efficacy. CRISPR/Cas9-based therapies are poised to transform cancer treatment, providing new possibilities for patients with malignancies that are difficult to treat as the field advances. Current clinical trials demonstrate the potential of gene-editing technologies to transform immunotherapy, with various innovative strategies being explored to enhance the precision and longevity of anticancer responses.

The emergence of CRISPR/Cas9 gene editing, along with breakthroughs in nanotechnology, signifies a transformative period in cancer therapy. Although several preclinical studies have demonstrated the impressive potential of this technology, its translation into clinical applications poses a considerable hurdle. Current clinical trials utilizing CRISPR/Cas9 technology are predominantly confined to ex vivo modifications, wherein cells such as T cells or stem cells are genetically altered externally before being reintroduced into the patient. Although intriguing, this method has some drawbacks. Conversely, in vivo gene editing presents unique benefits such as the capacity to simultaneously target different tissues and decrease expenses related to time and resources.

The clinical application of in vivo CRISPR/Cas9 gene editing is impeded by the limitations of existing delivery techniques. For therapeutic efficacy, nanomaterials must enable the effective intracellular distribution of CRISPR/Cas9 components, ensuring swift lysosomal escape and successful nuclear localization. Moreover, reducing the dimensions of Cas proteins is essential for their effective encapsulation into nanocarriers. Safety issues are of utmost importance in clinical translation. This requires temporary CRISPR/Cas9 expression and biocompatible nanomaterials. Mitigating the off-target consequences of CRISPR/Cas9 is essential. Strategies to improve specificity involve utilizing tissue-specific promoters and creating nanoparticles that allow spatiotemporal regulation and ligand-mediated active targeting. Compliance with the current Good Manufacturing Practice (CGMP) criteria is crucial for the clinical translation of nanocarriers.

### 5.1. Challenges and Future Outlook on Clinical Translation

While CRISPR-based therapies hold considerable promise, several challenges remain. Numerous trials have been terminated or withdrawn due to unforeseen complications, such as the trial targeting Wilms tumor 1 (WT1) using Cas9 RNP (NCT05066165) and the study involving PD-1 knockout T cells in metastatic RCC (NCT02867332). These setbacks underscore the need for continuous advancement of gene-editing technologies, which includes the development of more efficient delivery systems, minimization of off-target effects, and assurance of long-term safety. The advent of CRISPR/Cas9 gene editing, coupled with advancements in nanotechnology, marks a pivotal era in cancer treatment. While numerous preclinical studies have highlighted the remarkable potential of this technology, its application in clinical settings remains a significant challenge. Current clinical trials employing CRISPR/Cas9 technology are largely restricted to ex vivo modifications, wherein cells such as T cells or stem cells are genetically modified outside the body before being reintroduced into the patient. Although promising, this approach has certain limitations. In contrast, in vivo gene editing offers distinct advantages, including the ability to target multiple tissues simultaneously and reduce costs associated with time and resources.

The clinical implementation of in vivo CRISPR/Cas9 gene editing is significantly hindered by the constraints of current delivery methodologies. For therapeutic effectiveness, nanomaterials must facilitate the efficient intracellular distribution of CRISPR/Cas9 components, ensuring rapid lysosomal escape and successful nuclear localization. Additionally, minimizing the size of Cas proteins is crucial for their effective encapsulation within nanocarriers.

Addressing Off-Target Effects of CRISPR/Cas9: Mitigating the off-target consequences of CRISPR/Cas9 is crucial for its clinical application. Off-target effects, wherein unintended genomic sites are edited, can result in adverse outcomes, including genotoxicity, insertional mutagenesis, or undesired alterations in gene expression.

Strategies to enhance specificity include employing high-fidelity Cas9 variants (e.g., SpCas9-HF1, eSpCas9), optimizing guide RNA design, truncated sgRNAs, orthologous Cas systems, and developing nanoparticles that enable spatiotemporal regulation and ligand-mediated active targeting, thereby reducing systemic exposure and concentrating activity at the target site.

Additionally, the use of RNP-based delivery systems, which provide transient Cas9 expression, inherently diminishes the window for off-target activity compared to plasmid or viral vector delivery. Continuous advancements in in silico prediction tools for off-target sites and rigorous validation through sensitive molecular assays (e.g., CIRCLE-seq, GUIDE-seq, Digenome-seq) are essential tools for preclinical validation to minimize the risks in gene editing.

Ensuring Long-Term Biosafety of Nanocarriers: Safety considerations are paramount in clinical translation. For nanocarriers, this necessitates not only demonstrating acute safety but also ensuring long-term biosafety. This encompasses:

Biodegradability and Clearance: It is imperative that nanocarriers are engineered to be both biocompatible and biodegradable, thereby preventing their prolonged accumulation in tissues, which could result in chronic inflammation, toxicity, or compromised organ function.

Immunogenicity: Although nanocarriers generally exhibit lower immunogenicity compared to viral vectors, they can still elicit immune responses. A comprehensive evaluation of their potential to activate both innate and adaptive immunity is essential, particularly during initial administration and upon repeated dosing. This assessment should encompass the evaluation of cytokine storms, complement activation, and the formation of anti-nanocarrier antibodies.

Toxicity Profiles: Extensive preclinical toxicology studies are required to assess potential dose-dependent toxicities, including hepatotoxicity, nephrotoxicity, and systemic inflammatory responses. It is also crucial to demonstrate the safety of the degradation products of biodegradable nanocarriers.

Excipient Safety: A rigorous evaluation of the safety of all excipients and components used in the formulation of nanocarriers is necessary. Temporary CRISPR/Cas9 expression: Implementing delivery strategies that ensure transient CRISPR/Cas9 expression, such as those utilizing mRNA or RNP, is vital for enhancing safety by limiting the duration of gene editing activity. This approach reduces the risk of off-target effects and prevents persistent expression that could lead to undesirable cellular alterations.

Adherence to the current good manufacturing practice (CGMP) standards is essential for the clinical translation of nanocarriers, ensuring the consistent quality, purity, and safety of the final product. Current clinical trials underscore the transformative potential of gene-editing technologies in immunotherapy, with various innovative strategies being explored to enhance the precision and durability of anticancer responses. Despite the inherent complexities, ongoing advancements in nanocarrier design, coupled with a profound understanding of CRISPR/Cas9 biology and rigorous safety evaluations, indicate that these therapies are poised to revolutionize cancer treatment, offering new possibilities for patients with malignancies that are challenging to treat as the field progresses.

The effective clinical translation of CRISPR/Cas9-based cancer therapies is contingent on the development of safe and efficient delivery systems. As previously discussed, a range of nanocarrier platforms has been devised, each presenting unique advantages and limitations for the delivery of CRISPR/Cas9 components, whether in the form of plasmid DNA, mRNA, or ribonucleoprotein (RNP) complexes. To facilitate a systematic evaluation of these methodologies, we compiled their key characteristics in Table 1, emphasizing critical parameters such as biocompatibility, cellular uptake efficiency, endosomal escape capability, and gene editing performance, along with their primary benefits and limitations. This comparative analysis offers essential insights to guide future research and development of optimized nanocarriers tailored to specific cancer treatment applications. Ultimately, a balance between efficacy and safety, along with long-term follow-up studies, will be essential to realize the full clinical potential of CRISPR/Cas9 nanoparticle therapeutics in oncology.

### 5.2. Future Direction

The substantial potential of nanotechnology-enabled CRISPR/Cas9 delivery systems for cancer therapy is tempered by several fundamental challenges that necessitate innovative research. A critical priority is the development of next-generation nanocarriers with precisely engineered physicochemical and biological properties that are specifically designed to overcome organ-specific and tumor-type-specific barriers. In the context of glioblastoma and other intracranial malignancies, the formidable blood–brain barrier (BBB) requires multimodal strategies, including the following: (1) biomimetic nanovesicles functionalized with BBB-shuttling peptides (e.g., angiopep-2) or receptor-specific antibodies (anti-TfR and anti-LDLR) to exploit endogenous transport mechanisms; (2) transient physical disruption approaches that combine MRI-guided focused ultrasound with gas-filled nanovesicles for spatiotemporal control of BBB permeability; and (3) nose-to-brain delivery platforms utilizing mucoadhesive polymers and trigeminal nerve-targeting ligands. These strategies must be integrated with tumor-penetrating designs that exploit aberrant tumor vasculature while avoiding non-specific trapping by the reticuloendothelial system. Recent advancements in localized gene-editing delivery strategies have shown significant promise in both brain and pancreatic cancer models. In glioma treatment, focused ultrasound (FUS) coupled with microbubbles has been shown to transiently open the blood–brain barrier, allowing systemic CRISPR/Cas9 delivery with efficient tumor editing and minimal neural damage [87,88].

The distinct pathophysiology of pancreatic ductal adenocarcinoma poses unique challenges that require specialized interventions. Future research should focus on (1) stroma-modulating nanocarriers incorporating enzymatic payloads, such as PEGylated hyaluronidase and collagenase, or CAF-reprogramming CRISPR constructs targeting the TGF-β and SHH pathways; (2) dual-targeting systems with sequential ligand presentation, initially targeting the stroma followed by cancer cell-specific moieties; and (3) mechanistically engineered nanoparticles with optimized rigidity (elastic modulus approximately 10–50 kPa) and surface topology to enhance interstitial transport. For metastatic cancers, investigations should prioritize multi-organotropic delivery systems utilizing tropism-switching coatings responsive to organ-specific proteases or redox gradients, thereby enabling the sequential targeting of primary tumors and disseminated lesions. Similarly, microbubble-enhanced FUS has enabled the delivery of Cas9-sgRNA ribonucleoproteins into human induced pluoripotent stem cells and cancer cells. Targeted nanocarrier systems offer complementary innovations: anti-CD44-conjugated, olive oil-based nanocapsules selectively target cancer stem cells, enhancing precision therapeutics [89]; lipid-coated mesoporous silica nanoparticles loaded with irinotecan outperform liposomal delivery, showing higher efficacy and safety; and exosome-mediated CRISPR/Cas9 delivery directed at oncogenic KRAS^G12D mutations highlights a biologically compatible method to disrupt tumorigenesis [90]. Together, these breakthroughs suggest a future where AI-driven design of stimuli-responsive or tumor-targeted nanocarriers, combined with FUS-based or cell-derived delivery platforms, could enable precise, tumor-specific gene editing in otherwise hard-to-reach malignancies such as brain and pancreatic cancer.

The convergence of artificial intelligence and nanomedicine design marks a significant paradigm shift that warrants comprehensive exploration. Deep learning frameworks, including graph neural networks and transformer models, when trained on multi-omic datasets, can predict structure-function relationships among (1) nanocarrier physicochemical properties (such as hydrodynamic diameter, zeta potential, and PEG density), (2) biological interactions (such as protein corona formation and cellular uptake kinetics), and (3) therapeutic outcomes (such as gene editing efficiency and tumor regression). Reinforcement learning algorithms offer the potential to refine multi-objective design criteria, such as enhancing tumor delivery while reducing hepatic sequestration. Additionally, generative AI models can be used to suggest innovative biomaterials with specific degradation profiles. Successful application of these methodologies requires the development of standardized protocols for nanocarrier characterization and the creation of high-fidelity datasets that include in vitro, in vivo, and clinical data.

To achieve clinical translation, it is imperative to address key challenges in the following areas: (1) scalable manufacturing, which can be facilitated through quality-by-design (QbD) methodologies and inline analytical technologies (PAT); (2) development of regulatory science for combination nanomedicine/gene therapy products; and (3) health economics, particularly in terms of cost-effective production. The establishment of international consortia dedicated to standardized preclinical testing and data sharing is essential for expediting progress. By systematically tackling these multidisciplinary challenges through coordinated basic, translational, and clinical research, this field can harness the transformative potential of nanotechnology-powered CRISPR/Cas9 systems as precision therapeutics for treating intractable cancers.

## 6. Conclusions

Nanotechnology-based delivery systems offer a promising solution to the challenges of CRISPR/Cas9 delivery in cancer therapy. By addressing barriers, such as efficient encapsulation, targeted delivery, controlled release, and cellular internalization, nanocarriers enhance the therapeutic potential of CRISPR/Cas9. Continued research and development in this field are essential to translate these technologies into clinical applications. The integration of CRISPR/Cas9 with advanced nanocarriers has the potential to revolutionize cancer treatment, offering personalized and precise therapeutic options for patients.

## Figures and Tables

**Figure 1 cells-14-01136-f001:**
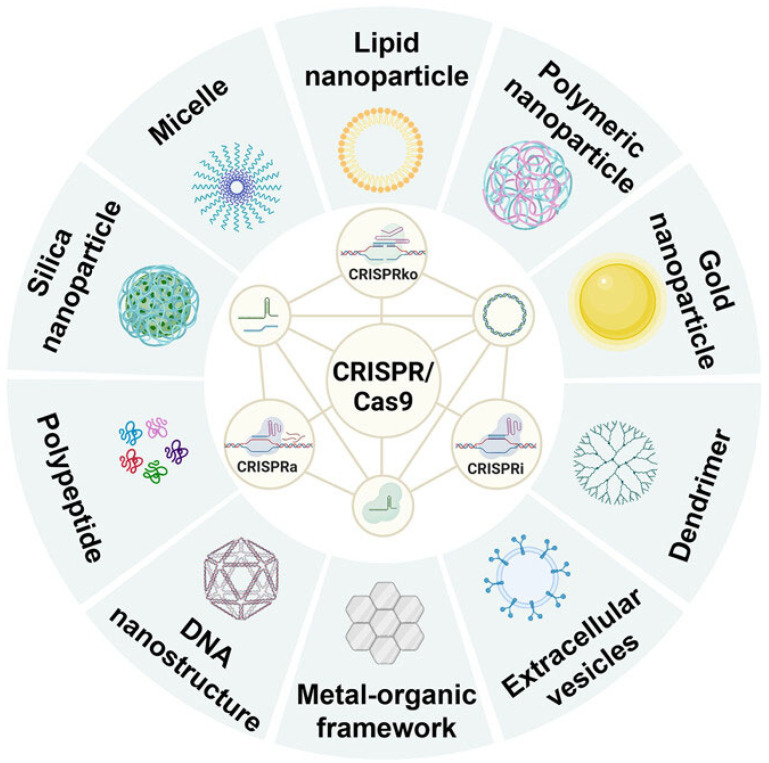
This figure provides a schematic overview of the CRISPR/Cas9 system and its delivery using nanotechnology. The CRISPR/Cas9 system, a powerful gene-editing tool, can be delivered in various forms, including DNA, mRNA/sgRNA, and ribonucleoprotein (RNP). This system enables diverse gene-editing approaches, such as CRISPR knockout (CRISPR KO), CRISPR interference (CRISPRi), and CRISPR activation (CRISPRa), each with unique functionalities. Various nanomaterials have been employed as carriers to effectively deliver the CRISPR/Cas9 system to target cells. These include lipid nanoparticles, polymers, inorganic compounds, polypeptides, dendrimers, and extracellular vesicles, each of which offers distinct advantages for efficient and targeted delivery. The figure is adopted from Zhou et al. 2024 [19] with permission.

**Figure 3 cells-14-01136-f003:**
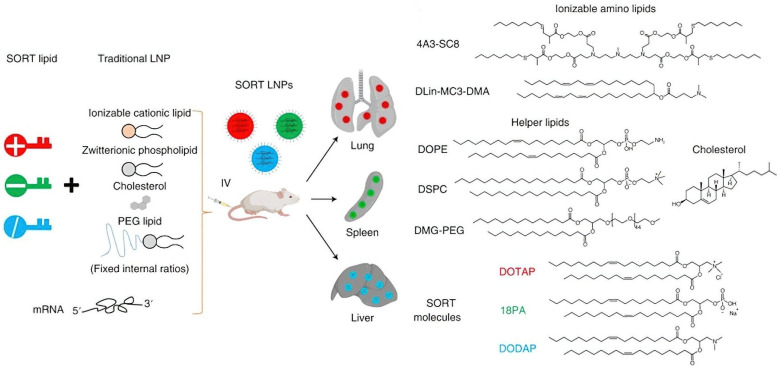
Selective organ targeting (SORT) lipid nanoparticles (LNPs) for mRNA delivery. Image adapted from Cheng et al. 2020 [71] with permission.

**Figure 4 cells-14-01136-f004:**
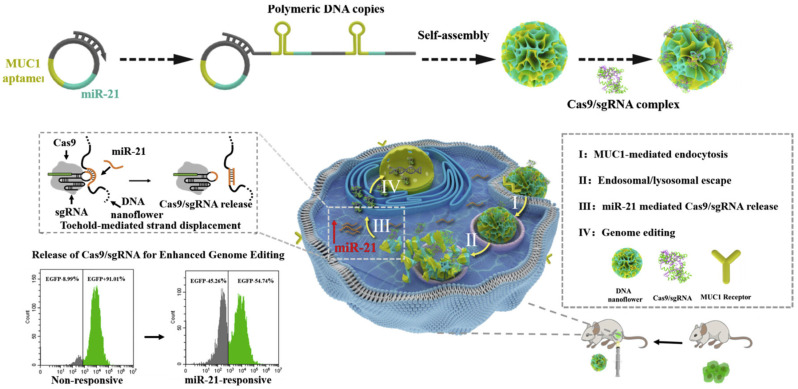
Schematic illustration of MUC1 aptamer-mediated delivery of an miR-21 responsive CRISPR/Cas9 system for enhanced genome editing. Adapted from Shi et al. 2020 [73].

**Figure 5 cells-14-01136-f005:**
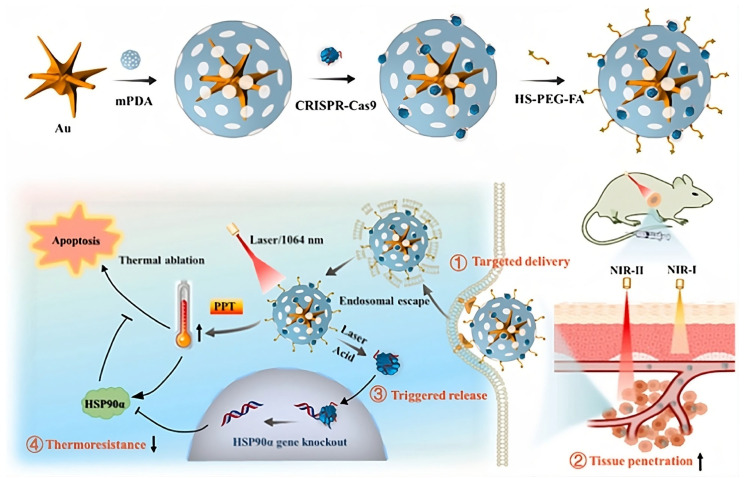
Schematic illustration of the fabrication process of CRISPR-Cas9@mPDA-PEG-FA nanoparticles and their mechanism of action in photothermal therapy. Figure adapted from Tao et al. 2021 [54] with permission.

**Figure 6 cells-14-01136-f006:**
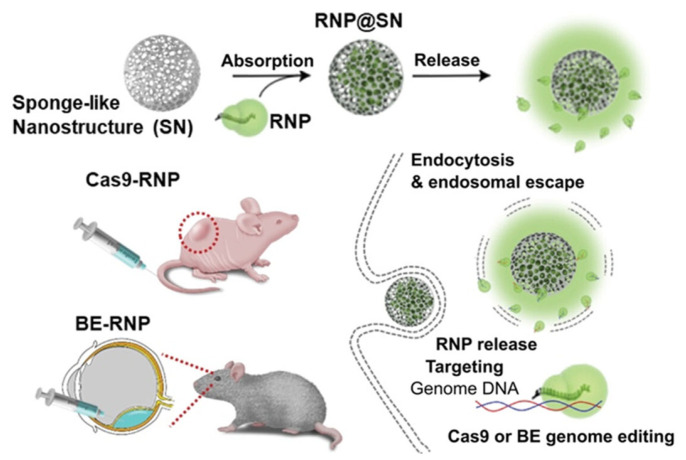
Schematic illustration of the targeted delivery and mechanism of action of Cas9 ribonucleoprotein (RNP)-loaded sponge-like nanostructures (SNs). Figure adapted from Kim et al. 2023 [80] with permission.

**Table 1 cells-14-01136-t001:** Comparative summary of nanocarriers for CRISPR/Cas9 delivery.

S.No.	Nanoparticles	Biocompatibility	Cellular Uptake/Endosomal Escape	Gene-Editing Efficiency	Key Advantages	Major Limitations
1	Lipid nanoparticles (LNP)	High	High	High	Proven clinical track-record (e.g., NTLA-2001); protects DNA; ligand-targetable	Possible immune activation; nuclear entry required; size-linked toxicity
2	LNPs-RNPs	High	High	Very high	Rapid editing; low off-target risk; >80% KO in vivo with SORT	Endosomal escape dose-limiting; off-target liver/spleen accumulation
3	mRNA + sgRNA (mainly LNP/iLNP)	High	Moderate–High	High	No nuclear step; fast; modified bases ↓ immunogenicity	mRNA instability and bulk; co-packaging challenge
4	Polymeric NPs (PBAE, dendrimer, chitosan)-plasmid	Moderate–High	Moderate–High	Moderate–High	Tunable charge/degradation; large payloads	Dose-dependent cytotoxicity; reproducibility
5	Polymeric NPs-RNP	High	High	High–Very high	Stable in blood; DNA-nanocages allow precise loading	Polymer-RNP stability and complement activation
6	Inorganic NPs (AuNP, CuS)	Moderate	High	High	Photothermal NIR release; tunable size/shape	Hepatic/splenic deposition; cost (Au)
7	MOF/ZIF	Moderate–High	Moderate	Moderate	Highly porous; stimuli-responsive; membrane-coatable	Early stage; potential metal-ion toxicity
8	Silica nanostructures/HMSN	High	Moderate–High	High	Huge loading; robust; TME-responsive	Complex surface chemistry; optimize escape
9	Extracellular vesicles (EVs)	Very high	Moderate	Moderate	Immune-silent; crosses BBB; intrinsic tropism	Low yield/loading heterogeneity; upscale cost
10	Polymeric micelles	High	Moderate	Moderate	Simple self-assembly; co-delivery friendly	Limited core size for RNP; premature unpacking

## Data Availability

Data sharing is not applicable.

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
