# Peer review of "Nanotechnology-Based Delivery of CRISPR/Cas9 for Cancer Treatment: A Comprehensive Review"

_cells, 2025, doi:10.3390/cells14151136_

Round 1
Reviewer 1 Report
Comments and Suggestions for Authors
- Although the content of the article is comprehensive, the description of certain parts (such as the specific preparation methods of nanocarriers) is relatively brief. It is suggested that the authors supplement some detailed steps or schematic diagrams of the preparation of key nanocarriers to help readers better understand the technical details.
- In the literature citations, there are minor inconsistencies in the citation format of some references (such as the abbreviation of journal names or the format of authors' names). It is recommended that the authors unify the citation format to comply with academic standards.
- The resolution of some figures is relatively low. It is recommended that the authors enhance the clarity of the figures to ensure they remain clear when enlarged. In addition, for some complex nanocarrier structures, it is suggested to add more labels and explanations to help readers better understand their design principles.
- When discussing clinical applications, it is suggested that the authors provide a more in-depth analysis of potential risks and challenges, such as how to address the off-target effects of the CRISPR/Cas9 system and how to ensure the long-term biosafety of nanocarriers, so that readers can have a more comprehensive understanding of the clinical application prospects of this technology.
- It is suggested that the authors provide a more in-depth analysis of the advantages and disadvantages of each nanocarrier research method, such as comparisons of different nanocarriers in terms of biocompatibility, cellular uptake efficiency, and gene editing efficiency, to help readers better assess the applicability of different methods.
- It is suggested that the authors further refine the discussion of future research directions, such as proposing more specific nanocarrier design strategies for specific types of cancer (such as brain tumors, pancreatic cancer, etc.), or exploring how to combine emerging technologies like artificial intelligence to further optimize the delivery efficiency of the CRISPR/Cas9 system.
- In some paragraphs, the sentence structure is relatively complex. It is suggested that the authors simplify the sentence structure appropriately to improve the readability of the article. In addition, it is recommended that the authors add some summarizing paragraphs in the article to briefly summarize the main content of each part, so that readers can better grasp the key points of the article.
- The understanding of nanomaterials in the introduction is not thorough. Also, the typical nanomaterials agents should be briefly listed/introduced in the first paragraph (one sentence should be sufficient). The authors refer to the literatures (10.1002/EXP.20230163; Oncology and Translational Medicine (2024) 10:4;151–161;).
Author Response
- Although the content of the article is comprehensive, the description of certain parts (such as the specific preparation methods of nanocarriers) is relatively brief. It is suggested that the authors supplement some detailed steps or schematic diagrams of the preparation of key nanocarriers to help readers better understand the technical details.
Response: Thank you reviewer, for your valuable comment we have modified the manuscript as per suggestion.
- In the literature citations, there are minor inconsistencies in the citation format of some references (such as the abbreviation of journal names or the format of authors' names). It is recommended that the authors unify the citation format to comply with academic standards.
Response: Thank you reviewer, for your valuable comment we have modified the manuscript as per suggestion.
- The resolution of some figures is relatively low. It is recommended that the authors enhance the clarity of the figures to ensure they remain clear when enlarged. In addition, for some complex nanocarrier structures, it is suggested to add more labels and explanations to help readers better understand their design principles.
Response: Thank you reviewer, for your valuable comment we have modified the manuscript as per suggestion and inserted new images
- When discussing clinical applications, it is suggested that the authors provide a more in-depth analysis of potential risks and challenges, such as how to address the off-target effects of the CRISPR/Cas9 system and how to ensure the long-term biosafety of nanocarriers, so that readers can have a more comprehensive understanding of the clinical application prospects of this technology.
Response: Thank you reviewer, for your valuable comment we have modified the manuscript as per suggestion.
- It is suggested that the authors provide a more in-depth analysis of the advantages and disadvantages of each nanocarrier research method, such as comparisons of different nanocarriers in terms of biocompatibility, cellular uptake efficiency, and gene editing efficiency, to help readers better assess the applicability of different methods.
Response: Thank you reviewer, for your valuable comment we have modified the manuscript as per suggestion.
- It is suggested that the authors further refine the discussion of future research directions, such as proposing more specific nanocarrier design strategies for specific types of cancer (such as brain tumors, pancreatic cancer, etc.), or exploring how to combine emerging technologies like artificial intelligence to further optimize the delivery efficiency of the CRISPR/Cas9 system.
Response: Thank you reviewer, for your valuable comment we have modified the manuscript as per suggestion.
- In some paragraphs, the sentence structure is relatively complex. It is suggested that the authors simplify the sentence structure appropriately to improve the readability of the article. In addition, it is recommended that the authors add some summarizing paragraphs in the article to briefly summarize the main content of each part, so that readers can better grasp the key points of the article.
Response: Thank you reviewer, for your valuable comment we have modified the manuscript as per suggestion.
- The understanding of nanomaterials in the introduction is not thorough. Also, the typical nanomaterials agents should be briefly listed/introduced in the first paragraph (one sentence should be sufficient). The authors refer to the literatures (10.1002/EXP.20230163; Oncology and Translational Medicine (2024) 10:4;151–161;).
Response: Thank you reviewer, for your valuable comment we have modified the manuscript as per suggestion.
Reviewer 2 Report
Comments and Suggestions for Authors
The review article by Rauf et al. describes the diverse nanoparticles for CRISPR/Cas-9 delivery.
-The manuscript is, in general, well-written. However, the abstract should be edited.
-It is not particularly novel because there are several review articles on this topic. Examples:
Seijas, A.; Cora, D.; Novo, M.; Al-Soufi, W.; Sánchez, L.; Arana, Á.J. CRISPR/Cas9 Delivery Systems to Enhance Gene Editing Efficiency. Int. J. Mol. Sci. 2025, 26, 4420. https://doi.org/10.3390/ijms26094420
Li, T., Yang, Y., Qi, H. et al. CRISPR/Cas9 therapeutics: progress and prospects. Sig Transduct Target Ther 8, 36 (2023). https://doi.org/10.1038/s41392-023-01309-7
Kim, M., Hwang, Y., Lim, S., Jang, H. K., & Kim, H. O. (2024). Advances in Nanoparticles as Non-Viral Vectors for Efficient Delivery of CRISPR/Cas9. Pharmaceutics, 16(9), 1197. https://doi.org/10.3390/pharmaceutics16091197.
-The manuscript describes twice each nanoparticle: once for mRNA or DNA Cas9/gRNA and other for RNP: Cas-9/gRNA. I think it is a better idea to describe each nanoparticle with its applications for DNA, mRNA, or protein: Cas9.
-The Figures are mislabeled. There are two Figure 4. The first Figure 4 is not relevant to this topic because it is related to siRNA delivery.
-The resolution of the other Figures is low and should be improved
- Although the topic is cancer, at some point it is important to mention the CRISPRs currently in the clinic: Casgevy and carbamoyl phosphate synthetase 1 (CPS1) deficiency.
-Figure 1 is too simple and it does not explain how to supply a DNA oligo to replace a mutated DNA for therapy: “cut and paste”.
-Line 156: Additional modifications with DSPE-PEG-HA.. should be expanded to clarify
-Lines 365-372: ..xenograft tumor model… explain better: what cancer type, where the tumor was located, etc.
Other minor points:
Line 19: … investigates.. better: revises
Line 41:…formation…better: initiation
Line 87:…simple.. better: ancient
Line 279: ..transpires…better: occurs
Line 322:…technology… better: approach
Author Response
he review article by Rauf et al. describes the diverse nanoparticles for CRISPR/Cas-9 delivery.
-The manuscript is, in general, well-written. However, the abstract should be edited.
-It is not particularly novel because there are several review articles on this topic. Examples:
Seijas, A.; Cora, D.; Novo, M.; Al-Soufi, W.; Sánchez, L.; Arana, Á.J. CRISPR/Cas9 Delivery Systems to Enhance Gene Editing Efficiency. Int. J. Mol. Sci. 2025, 26, 4420. https://doi.org/10.3390/ijms26094420
Li, T., Yang, Y., Qi, H. et al. CRISPR/Cas9 therapeutics: progress and prospects. Sig Transduct Target Ther 8, 36 (2023). https://doi.org/10.1038/s41392-023-01309-7
Kim, M., Hwang, Y., Lim, S., Jang, H. K., & Kim, H. O. (2024). Advances in Nanoparticles as Non-Viral Vectors for Efficient Delivery of CRISPR/Cas9. Pharmaceutics, 16(9), 1197. https://doi.org/10.3390/pharmaceutics16091197.
-The manuscript describes twice each nanoparticle: once for mRNA or DNA Cas9/gRNA and other for RNP: Cas-9/gRNA. I think it is a better idea to describe each nanoparticle with its applications for DNA, mRNA, or protein: Cas9.
-The Figures are mislabeled. There are two Figure 4. The first Figure 4 is not relevant to this topic because it is related to siRNA delivery.
-The resolution of the other Figures is low and should be improved
- Although the topic is cancer, at some point it is important to mention the CRISPRs currently in the clinic: Casgevy and carbamoyl phosphate synthetase 1 (CPS1) deficiency.
-Figure 1 is too simple and it does not explain how to supply a DNA oligo to replace a mutated DNA for therapy: “cut and paste”.
-Line 156: Additional modifications with DSPE-PEG-HA.. should be expanded to clarify
-Lines 365-372: ..xenograft tumor model… explain better: what cancer type, where the tumor was located, etc.
Other minor points:
Line 19: … investigates.. better: revises
Line 41:…formation…better: initiation
Line 87:…simple.. better: ancient
Line 279: ..transpires…better: occurs
Line 322:…technology… better: approach
Response: We have worked on the said comments and modifications have been done in the modified manuscript.
Round 2
Reviewer 2 Report
Comments and Suggestions for Authors
Most of the critics raised during the first revision were properly addressed
The number of the figures are still mislabeled.